# Non-Invasive Lung Cancer Diagnostics through Metabolites in Exhaled Breath: Influence of the Disease Variability and Comorbidities

**DOI:** 10.3390/metabo13020203

**Published:** 2023-01-30

**Authors:** Azamat Z. Temerdashev, Elina M. Gashimova, Vladimir A. Porkhanov, Igor S. Polyakov, Dmitry V. Perunov, Ekaterina V. Dmitrieva

**Affiliations:** 1Department of Analytical Chemistry, Kuban State University, Stavropol’skaya St. 149, Krasnodar 350040, Russia; 2Research Institute–Regional Clinical Hospital N° 1 n.a. Prof. S.V. Ochapovsky, 1 May St. 167, Krasnodar 350086, Russia

**Keywords:** volatile organic compounds, exhaled breath, lung cancer, thermal desorption, GC-MS, comorbidities

## Abstract

Non-invasive, simple, and fast tests for lung cancer diagnostics are one of the urgent needs for clinical practice. The work describes the results of exhaled breath analysis of 112 lung cancer patients and 120 healthy individuals using gas chromatography-mass spectrometry (GC-MS). Volatile organic compound (VOC) peak areas and their ratios were considered for data analysis. VOC profiles of patients with various histological types, tumor localization, TNM stage, and treatment status were considered. The effect of non-pulmonary comorbidities (chronic heart failure, hypertension, anemia, acute cerebrovascular accident, obesity, diabetes) on exhaled breath composition of lung cancer patients was studied for the first time. Significant correlations between some VOC peak areas and their ratios and these factors were found. Diagnostic models were created using gradient boosted decision trees (GBDT) and artificial neural network (ANN). The performance of developed models was compared. ANN model was the most accurate: 82–88% sensitivity and 80–86% specificity on the test data.

## 1. Introduction

The development of new non-invasive and comfortable methods to diagnose various diseases is an urgent task in modern medicine. Exhaled breath [1], exhaled breath condensate [2], saliva [1,3], skin [1,4,5], and urine [1,6] are intensively studied to develop new diagnostic approaches. Exhaled breath is especially interesting for diagnostic purposes since it can be obtained without any discomfort for patients [7].

A few non-invasive tests have already been implemented in clinical practice: 13C-urea breath test in the diagnostics of Helicobacter pylori infection [8], nitric oxide breath test in asthma, and allergic airway inflammation management [9]. However, many diseases with high mortality rate are still diagnosed using complex and invasive procedures. Lung cancer remains the leading cause of death [10], since the disease develops rapidly and asymptomatically at the initial stage and can be diagnosed only by harmful and invasive procedures such as low dose computed tomography (LDCT) and biopsy. Biopsy is an invasive procedure; LDCT scanning includes radiation exposure. As such, the development of new, accurate, simple to use and non-invasive methods for lung cancer diagnostics is highly required.

Lung cancer biomarkers can be identified using various analytical methods [11,12]. Among them, gas chromatography coupled with mass spectrometry (GC-MS) is extremely useful since it allows to conduct quantitative and qualitative analysis of the samples. Many scientists reported results of exhaled breath analysis using GC-MS [13,14,15,16,17] to identify lung cancer biomarkers. Multidimensional gas chromatography seems to be particularly useful to consider such complex issues as biomarkers identification [13].

GC-MS is a laborious method which demands extensive experience. Proton transfer reaction mass spectrometry (PTR-MS) [18], selected ion flow tube mass spectrometry (SIFT MS) [19], ion mobility spectrometry (IMS) [20] are also applied to solve the task. They allow to perform fast analysis of exhaled breath without any sample preparation.

Another possible analytical scheme is obtaining the informative signal from the whole exhaled breath composition instead of searching for specific biomarkers. Electronic noses based on different sensor systems are vigorously applied to address the issue. The most popular electronic noses are Aeonose^®^ [21] and Cyranose 320 [22] which are commercially available. They were used to analyze exhaled breath for lung cancer diagnostics. There exist electronic noses based on a plethora of different sensor types such as colorimetric sensors [23], nanomaterials [24], quartz crystal microbalance sensors [25,26], and combined gas sensors [27] for exhaled breath analysis.

It is hardly possible to diagnose lung cancer by a unique marker; no biomarker separately can diagnose the disease accurately enough to be applied in clinical practice. Mostly, the discrimination of lung cancer patients and healthy controls can be achieved applying statistical data analysis methods. As a rule, Wilcoxon rank sum test [28,29] or Mann–Whitney U-test [18] are applied to identify statistically significant differences between exhaled breath samples of the investigated cohorts of people. Different machine learning algorithms, such as support vector machine [22], k-nearest neighbor classifier [27], and others [13,16,19,21,30,31], have been applied to create diagnostic models.

The results obtained by different research groups in the field of exhaled breath analysis for lung cancer diagnostics using various analytical methods are significantly different. No consistency in the set of biomarkers, statistical data analysis methods, and performance of predictive models is observed [32]. Traditionally, the group of lung cancer patients includes patients with various exact diagnoses. The resulting products of metabolomic pathways excreted in exhaled breath can vary significantly for different kinds of malignancy. It can contribute to inconsistency of the results. On the other hand, patients with lung cancer can suffer from other diseases, which can also affect the results. To find lung cancer biomarkers, the scientists prefer to exclude the patients with other lung diseases. However, not only lung comorbidities, but other pathologies can alter VOC profile, such as diabetes, hypertension, and so on. Several researchers have presented the results of exhaled breath analysis for diagnosing other diseases: diabetes [33], obesity [34], heart failure [35], and others. However, variation in VOC profile of lung cancer patients dependently from other non-pulmonary comorbidities has not been studied.

This research is devoted to the study of lung cancer variability and comorbidities. Effects of chronic heart failure, hypertension, anemia, acute cerebrovascular accident, obesity, and diabetes on VOC profile of lung cancer patients were studied. The influence of the most frequently occurring comorbidities among lung cancer patients was investigated for the first time. TD-GC-MS was used to analyze exhaled breath samples of lung cancer patients and healthy volunteers. VOC peak areas and their ratios were considered as quantitative parameters. Additionally, variation in exhaled breath composition of lung cancer patients dependently from tumor histological type, localization, TNM stage, and effect of chemotherapy was studied. The parameters, which varied dependently from treatment status and comorbidities, were not considered as putative lung cancer biomarkers. Two kinds of machine learning algorithms, namely, gradient boosted decision trees (GBDT) and artificial neural network (ANN), were applied for the creation of a diagnostic model.

## 2. Materials and Methods

### 2.1. Materials

Ethyl ether (>95%) was obtained from Acros Organics (New Jersey, USA). Benzene, toluene, n-hexane, acetonitrile, methanol, and ethanol (>95%) were purchased from Sigma–Aldrich (St. Louis, MI, USA). N-butanol, 2-butanol, and acetone (99.9%) were obtained from Ecos-1 (Moscow, Russia); 2-propanol was obtained from Vecton (Moscow, Russia). Butyl acetate and ethyl acetate and (99%) were purchased from Component-Reaktiv (Moscow, Russia).

### 2.2. Human Subjects

The study involved 2 groups of participants: lung cancer patients and healthy volunteers. A volunteer was defined as healthy based on a yearly physical exam report. Inclusion criteria were absence of pathologies and inflammation processes in lungs, which was verified by fluorography. Diagnosis of lung cancer patients was confirmed by biopsy. Patients with other lung comorbidities along with lung cancer were excluded. Most patients were treated with chemotherapy (88 patients), immunotherapy (7 patients), or target therapy (1 patient). The rest individuals provided the samples before a treatment course. Information on the volunteers is reflected in Table 1. Each participant provided an informed consent.

### 2.3. Exhaled Breath Collection

Mixed expiratory breath samples were collected in 5-L Tedlar (Supelco, Bellefonte, PA, USA) sampling bags pre-cleaned by flushing with nitrogen. The possibility of sample pollution by compounds from the sampling bag was studied earlier [36]. The intensities of phenol and *N*,*N*-dimethylacetamide increased after 2 h of sample storage in the sampling bag. Therefore, these compounds were omitted from a list of putative biomarkers. The samples of lung cancer patients and some healthy volunteers were collected in the hospital. The samples of other healthy volunteers were collected in a room without solvents. Ambient air was sampled on the day of exhaled breath sampling to consider the influence of exogenous compounds. The subjects were fasted overnight before breath sampling. Exhaled breath of active smokers was sampled not earlier than 2.5 h after smoking. It was found that anatomic dead space, breath hold, and flow rate might affect the results in case of healthy volunteers but not in lung cancer patients [37]. Therefore, it was essential to provide the same sampling conditions for both groups of participants. On the other hand, establishing the certain flow rate during sampling can be associated with discomfort and pain for patients. Therefore, these parameters were not controlled. However, the sampling procedure was the same for both cohorts of people. It was conducted as follows: after a 10-min rest in a sampling room, volunteers were asked to deeply breathe, hold their breath for 10 s and breathe out into the sampling bag in a calm manner, repeating the procedure until filling it. Breath samples were stored in sampling bags no longer than 6 h after sampling.

### 2.4. GC-MS Analysis of Exhaled Breath

The samples of exhaled breath were analyzed by GC-MS. A system consisting of a gas chromatograph (Chromatec crystal 5000.2, Yoshkar-Ola, Russia) coupled with a quadrupole mass spectrometer equipped with an electron ionization source (Chromatec MSD, Yoshkar-Ola, Russia), combined with a two-stage thermal desorber TD2 (Chromatec, Yoshkar-Ola, Russia). The Chromatec Analytic (Chromatec, Yoshkar-Ola, Russia) software and the mass spectral library NIST 2017, Version 2.3 (Gatesburg, PA, USA) were used for data acquisition and processing. Sorbent tubes with the external diameter and length of 6.2 and 115 mm filled with 0.4 g of Tenax TA (60–80 mesh, Chromatec, Yoshkar-Ola, Russia) sorbent with the surface area of 35 m^2^/g were used to preconcentrate VOCs. Exhaled breath VOCs were preconcentrated upon passing a 0.5-L sample through a Tenax TA sorbent tube at a rate of 200 mL/min using a PV-2 aspirator (Chromatec, Yoshkar-Ola, Russia). Supelco Supel-Q PLOT (30 m × 0.32 mm × 15 μm) column was applied to separate the compounds. The flow rate of carrier gas was 1.30 mL/min. Oven temperature program was as follows: initial 50 °C ramped at 10 °C/min to 150 °C, next ramped at 6 °C/min to 220 °C and finally ramped at 4 °C/min to 250 °C. GC-MS analysis conditions were optimized earlier (Table 2) [36]. Identification of VOCs was conducted by applying analytical standards by introducing gaseous compounds in the sorbent tube with subsequent thermal desorption and GC-MS analysis. The rest VOCs were identified by comparing the obtained mass spectra with library ones. All VOCs showing mass spectra with match factor ≥ 85% were considered.

### 2.5. Statistical Analysis

The chromatograms of exhaled breath samples were recorded in the full scan mode for quantification purposes. Extracted ion chromatogram (EIC) mode was applied to calculate the peak areas. The room air influence was eliminated by subtraction of room air peak area values from the exhaled breath. Negative results were equated to zero. To provide the reliability of the results, statistical analysis was conducted only for VOCs with peak areas at least 20% greater than in ambient air and occurring in more than 50% of samples. The ratios of the compound peak areas to the main ones (more than 86% of the samples) as well as ratios of the main VOCs were considered for statistical analysis.

The statistical analysis was conducted using StatSoft STATISTICA (version 10). Preliminary sample size calculations were conducted using power analysis to determine the minimum sample size required. For the correlation analysis, the results showed that the required sample size to achieve 85% power for detecting a correlation at a level of 0.2 at a significance criterion of α = 0.05 was N = 221. The study includes 232 samples, which is considered adequate. The normality of distribution was estimated using Kolmogorov–Smirnov test. Due to the non-normal distribution, nonparametric Spearman’s rank correlation test (*p* = 0.05) was applied to identify statistically significant correlations between the peak areas of VOCs, their ratios and disease status.

Spearman’s rank correlation test (*p* = 0.05) was used to find statistically significant correlations between the parameters and tumor localization (central or peripheral). Histological tumor type groups (adenocarcinoma, squamous cell carcinoma, and small cell carcinoma) were ranged according to their malignant course, from least malignant to the highest, in the following order: squamous cell carcinoma, adenocarcinoma, and small cell carcinoma. The correlation analysis was applied to estimate correlation between the parameters and malignant course as well as TNM stage.

The influence of chemotherapy on VOC profile was evaluated by comparing the VOC profiles of patients before beginning of any treatment and under the treatment using the correlation analysis. Further, the correlation analysis was used to estimate the effect of comorbidities on exhaled breath VOC profile.

The dataset was randomly divided into 2 datasets: training (70%) and test (30%) to create a diagnostic model. Sensitivity and specificity for both training and test data were calculated for each model. Gradient boosted decision trees (GBDT) and artificial neural network (ANN) were applied for the creation of a diagnostic model. ANN was trained using Broyden–Fletcher–Goldfarb–Shanno algorithm. Multilayer perceptron artificial neural network with one hidden layer was used to create the diagnostic model. The hidden layer included 5 neurons and the output layer contained 2 neurons, which determined whether the input data belonged to the healthy or lung cancer group.

## 3. Results

Exhaled breath samples of 112 lung cancer patients and 120 healthy volunteers were analyzed by GC-MS. Typical GC-MS chromatograms of exhaled breath samples from a lung cancer patient and a healthy volunteer are shown in Figure 1. A total of 205 VOCs were identified in the study. Table 3 represents the most frequently occurring compounds. The VOCs occurring in more than 50% of samples were used for statistical analysis.

Statistical analysis was conducted using VOC peak areas and their ratios. In case of ratios, to avoid division by zero, it is reasonable to use VOCs with frequency of occurring of 100% as a denominator, which was observed for acetone, isoprene, and dimethyl sulfide (Table 3). To consider a wider list of ratios, it was rational to apply the VOCs occurring the most frequently in the samples of both groups as a denominator, which was observed for the first 10 VOCs (Table 3). Among them, the lowest frequency was observed for acetonitrile. The frequency of occurring for rest compounds was lower and was different in the studied groups. These VOCs were applied only as a numerator.

At the initial step of the study, the correlation coefficients between VOC peak areas, their ratios and different factors were evaluated. Several ratios of 2-heptanone significantly correlated with the treatment status (before or under chemotherapy course): 2-heptanone/1-methylthiopropane, 2-heptanone/1-methylthiopropene, and 2-heptanone/dimethyl disulfide (correlation coefficients of −0.196, −0.206, −0.202, respectively).

The influence of different comorbidities on exhaled breath VOC profile of lung cancer patients was estimated using the correlation analysis. Statistically significant correlation between such comorbidities as anemia and acute cerebrovascular accident status, and VOC peak areas and their ratios was not found. Benzaldehyde/acetonitrile and benzaldehyde/2.3-butandione ratios significantly correlated with obesity status (correlation coefficients of 0.237 and 0.240). 2-Butanone (0.245) and some ratios of benzaldehyde and 2-butanone, i.e., benzaldehyde/allyl methyl sulfide (0.211), benzaldehyde/1-methylthiopropene (0.212), benzaldehyde/dimethyl sulfide (0.230), significantly correlated with diabetes status. Toluene and some ratios significantly correlated with both chronic heart failure and hypertension: toluene (−0.220 and −0.268), toluene/acetonitrile (−0.196 and −0.237), toluene/isoprene (−0.214 and −0.257), toluene/1-methylthiopropene (−0.208 and −0.257), toluene/dimethyl disulfide (−0.252 and −0.270), and 1-pentanol/dimethyl disulfide (−0.213 and −0.215). Significant correlation only with hypertension was observed for 1-pentanol (−0.205), toluene/allyl methyl sulfide (−0.236), 1-pentanol/2-butanone (−0.214), 1-pentanol/2,3-butandione (−0.218), and pentanal/acetone (−0.202).

In case of tumor localization, statistically significant correlations were found for 1-pentanol (correlation coefficient of 0.222), 1-pentanol/2,3-butandione (0.262), 1-pentanol/isoprene (0.210), 1-pentanol/acetone (0.193), dimethyl disulfide/acetonitrile (0.196), and 2-butanone/isoprene (0.191).

Statistically significant correlations were found between TNM stage and some VOC peak areas and their ratios (Table 4).

In case of tumor histological type ranged by malignant course, statistically significant correlations were found only for some VOC peak area ratios (Table 5).

Further, the correlation analysis was applied to find difference between the parameters of lung cancer patients and healthy individuals. Significant correlations between the disease status and peak areas of several VOC were found, i.e., acetone (−0.163), 1-methylthiopropene (0.140), 2-pentanone (0.244), hexane (−0.287), toluene (0.249), pentanal (−0.254), and dimethyl trisulfide (0.260). Additionally, a lot of ratios were significantly different between lung cancer patients and healthy volunteers. The ratios with the highest correlation coefficients were selected for the development of diagnostic models (Table 6). The group of healthy volunteers was significantly younger than lung cancer patients. To avoid confounding influence of age, correlation between age and all ratios selected for the creation of diagnostic models was estimated in groups of lung cancer patients and healthy volunteers separately (Table 6). None of ratios selected for the creation of diagnostic models had statistically significant correlation with age.

Diagnostic models were created using GBDT and ANN. The input values of each model represented the same set of 12 ratios (Table 6). To provide reliability of diagnostic models, 3-fold cross validation method was applied. Performance of models created using 3 datasets is shown in Table 7. The highest sensitivity on the training dataset was observed in case of GBDT. However, ANN diagnostic model has the best accuracy on the test dataset.

GBDT allows to estimate the importance of all variables which construct the model in relation to the most important ones. Bar plots illustrate the importance of the variables for each dataset (Figure 2). As it can be seen, the ratio of hexane/2-pentanone contributes less to the prediction in all datasets, but ratios of dimethyl trisulfide/dimethyl disulfide and isoprene/acetone were the most significant for distinguishing the two groups.

## 4. Discussion

Different research groups have shown an ability of exhaled breath analysis to diagnose lung cancer [17,18,19,22,31]. However, the results obtained by these groups were incoherent. Numerous analysis conditions, groups of volunteers, putative biomarker sets were used for the creation of diagnostic models, various learning algorithms and different performances of the models were obtained.

Inconformity of the results partially can be explained by variability of lung cancer groups in different studies. One of the aims of this study was to evaluate possible VOC profile variations in lung cancer group dependently from different factors. Considering that a part of lung cancer patients involved in the study was under the treatment, it was essential to evaluate the influence of treatment on exhaled breath composition. For this, correlations between the treatment status (before or under chemotherapy course) and VOC profile were calculated. Several ratios of 2-heptanone significantly correlated with the treatment status. The majority of other research groups have studied the effect of treatment only in case of surgery [16,38]. The results of exhaled breath analysis for monitoring response to treatment in lung cancer were demonstrated in work [39]. The effect of different kinds of treatment was studied. Alterations in concentrations of dodecane, styrene, 4-methyldodecane, and α-phellandrene were observed after treatment. We did not consider these VOCs due to low frequency of occurring in samples. In our study, the majority of lung cancer patients were under the treatment. Therefore, it was essential to exclude the VOCs and ratios affected by the treatment status. It was found that a small number of ratios were affected by the treatment status. However, the issue should be considered in detail in further studies.

Another issue which can influence the results is comorbidities. The better part of scientists prefers to exclude patients with lung comorbidities when it comes to involving volunteers to the study. We also excluded patients with other lung comorbidities in the study. However, metabolic pathway of other pathologies can also lead to alterations in exhaled breath profile. In this study, for the first time, the effect of non-pulmonary comorbidities on exhaled breath profile of lung cancer patients was studied. Hypertension and diabetes affect the VOC profile the most. The parameters correlating with other pathologies should be excluded to avoid their effect on discrimination of lung cancer patients and heathy volunteers. The current study has several limitations considering comorbidities. First, we have not studied the effect of other lung comorbidities, which can influence other parameters. Second, not all possible comorbidities were considered. Third, the ratio of the patients with comorbidities was low compared with patients without them. Therefore, the list of parameters correlating with other comorbidities can be extended with increasing the cohort of participants. However, the analysis of comorbidities effect allows us to exclude parameters correlating with other diseases and eliminate their influence.

Exhaled breath composition can be varied dependently of tumor localization. A lung tumor localized in the central part is closer to the airways than peripheral, which can affect the alterations in VOC profile of patients differently. Further, 1-pentanol and some its ratios as well as dimethyl disulfide/acetonitrile and 2-butanone/isoprene significantly correlated with tumor localization. Differences in VOC profiles of lung cancer patients in relation to tumor localization have never been investigated by other researchers. This issue should be considered further to confirm the results.

In case of TNM stage, peak areas of 2-butanone, 1-methylthiopropene and some ratios (Table 4) significantly correlated with TNM stage. Concentration of 2-butanone was found to be higher in advanced stages [29], which was proved by our findings, since positive significant correlation with TNM stage was observed. However, in other studies, no differences were found in VOC profiles of patients with different stages of lung cancer [15,29,40]. In this study, VOC profiles were considered based on the detailed TNM diagnosis instead of lung cancer stage (I, II, III, IV), which allows to reveal common tendencies not unclear when considering the groups separately. However, conformity with other works was observed, which proves the effect of disease stage on 2-butanone levels.

The results of different research groups concerning histological type effect are inconsistent. In a study [15], no differences were found in exhaled breath composition of patients with different histological types. Statistically significant differences in 1-butanol and 3-hydroxy-2-butanone concentrations in samples of patients with adenocarcinoma and squamous cell carcinoma were found in a study [29]. We have studied VOC profiles of patients with different histological types dependently from tumor malignancy (squamous cell carcinoma, adenocarcinoma, and small cell carcinoma). Statistical analysis has shown that no VOC significantly correlated with the tumor histological type. However, predominantly dimethyl trisulfide and 3-heptanone ratios (Table 5) significantly correlated with the histological type. Considering the inconformity of the results obtained by different research groups, it is worthy to continue the study involving larger cohorts of participants.

Previously, we have optimized analysis conditions, proposed a new approach for the data analysis by using VOC ratios instead of VOC peak area values and demonstrated the efficiency of the proposed approach using different analytical methods (GC-FID and GC-MS) and different cohorts of participants [36,41]. In this work, we analyzed exhaled breath of a significantly larger cohort of volunteers by GC-MS. A wider range of VOC ratios was considered for the creation of diagnostic models as well. We used GBDT to estimate the contribution of ratios in predictive power of the model. Dimethyl trisulfide/dimethyl disulfide ratio contributes the most in classification of the groups. The same results were obtained earlier on a lower cohort of people [41], which confirms reliability of the ratio as a lung cancer biomarker. ANN outperform GBDT in terms of performance on the test dataset with 82–88% sensitivity and 80–86% specificity. The performance of the previous model [36] was higher (more than 90% for both sensitivity and specificity), which can be caused by several reasons: first, the larger cohort of people was involved in the present research and 30% of samples instead of 15% were assigned to the test dataset, which increases the reliability of the present models; second, in this research, the participants were fasted overnight before sampling, which can additionally decrease interfering effects; third, the number of smoking participants was equal; thus, the influence of smoking was eliminated. Most lung cancer patients are active smokers, which does not allow us to consider only the patients who do not smoke. However, cigarette smoking significantly influences exhaled breath composition [42]. Thus, to consider smoking factor we had to involve a comparable number of smokers in both lung cancer patient and healthy volunteer groups. The group of healthy volunteers was significantly younger, than lung cancer patients. However, none of parameters selected for the creation of diagnostic models had statistically significant correlation with age. Further, it should be noted that the young participants (from 21 years old) were also present in the group of lung cancer patients. Unfortunately, the increasing trend in the number of young people among cancer patients should be considered.

Notably, significant correlations with the disease status were observed for ratios of toluene/acetonitrile, hexane/acetonitrile, and pentanal/isoprene. The same results were obtained in the previous research (correlation coefficients of 0.248, −0.307, −0.296, respectively). It evidences the robustness of the parameters as potential lung cancer biomarkers. Further study is required to confirm the reliance of other biomarkers found in this study.

## 5. Conclusions

The study has revealed that not only pulmonary, but also non-pulmonary comorbidities can influence the exhaled breath VOC profile. Among them, chronic heart failure and hypertension affect mostly. Variations in exhaled breath VOC profiles among lung cancer patients with different histological types, TNM stages, tumor localization, and treatment status have been observed, which can influence the performance of diagnostic models. These factors should be considered before creating a lung cancer diagnostic model, which allows to build a useful test for diagnosing the dangerous disease timely.

## Figures and Tables

**Figure 1 metabolites-13-00203-f001:**
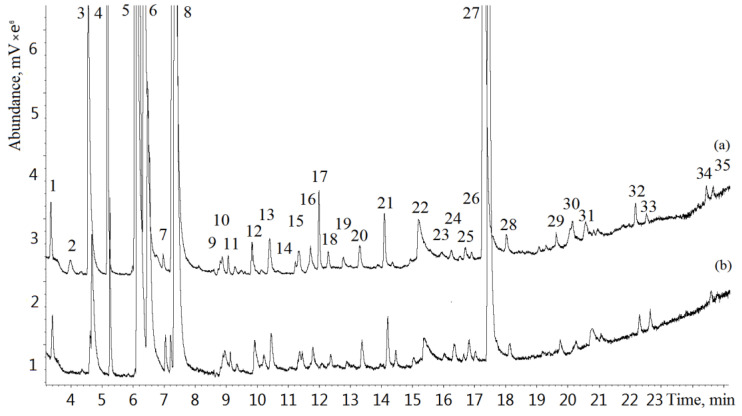
Typical total ion current (TIC) chromatograms of a lung cancer patient (a) and a healthy individual (b): 1—acetaldehyde, 2—isobutene, 3—ethanol, 4—acetonitrile, 5—acetone, 6—2-propanol, 7—ethyl ester, 8—isoprene, 9—2,3-butandione, 10—2-butanone, 11—dimethyl carbonate, 12—hexane, 13—benzene, 14—2-pentanone, 15—allyl methyl sulfide, 16—1-methylthiopropane, 17—1-methylthiopropene, 18—heptane, 19—1-pentanol, 20—toluene, 21—hexanal, 22—*N*,*N*-dimethylacetamide, 23—ethylbenzene, 24—o-xylene, 25—2-heptanone, 26—heptanal, 27—phenol, 28—benzaldehyde, 29—octanal, 30—decane, 31—limonene, 32—nonanal, 33—undecane, 34—decanal, 35—dodecane.

**Figure 2 metabolites-13-00203-f002:**
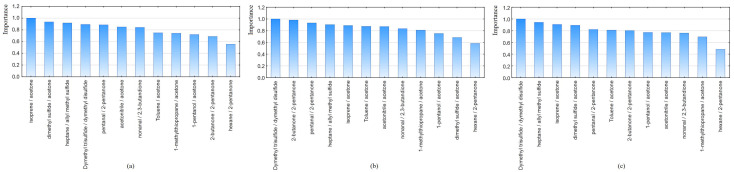
Importance bar plots of variables using GBDT, created on the datasets (**a**) 1, (**b**) 2, (**c**) 3.

**Table 1 metabolites-13-00203-t001:** Clinical characteristics of subjects.

Cohort	Feature	Number
Healthy participant	Number	120
	Male	36
	Female	84
	Age (median, range)	21, 21–67
	Smokers	17
Lung cancer	Number	112
	Male	88
	Female	24
	Age (median, range)	63, 21–77
	Smokers	22
	Localization of tumor	
	Central	59
	Peripheral	53
	Histology	
	Adenocarcinoma	50
	Squamous cell carcinoma	38
	Small cell carcinoma	12
	Non differentiated	12
	TNM (tumor, nodules, metastasis) stage	
	T1N0M1	1
	T2N0M0	8
	T2N0M1	5
	T2N1M0	8
	T2N1M1	1
	T2N2M0	2
	T2N2M1	6
	T2N3M0	1
	T2N3M1	2
	T3N0M0	6
	T3N0M1	2
	T3N1M0	7
	T3N1M1	1
	T3N2M0	15
	T3N2M1	4
	T3N3M0	1
	T4N0M0	5
	T4N0M1	5
	T4N1M0	8
	T4N1M1	1
	T4N2M0	10
	T4N2M1	10
	T4N3M0	2
	T4N3M1	1
	Comorbidity	
	Chronic heart failure	19
	Hypertension	18
	Anemia	8
	Acute cerebrovascular accident	5
	Obesity	4
	Diabetes	4

**Table 2 metabolites-13-00203-t002:** Thermal desorber and GC-MS operation modes.

Equipment	Parameter	Value
Thermal desorber	Carrier gas	Helium
	Carries gas flow rate (desorption from the sorption tube), mL/min	30
	Valve temperature, °C	150
	Transition line temperature, °C	180
	Desorption temperature, °C	250
	Initial trap temperature, °C	−10
	Final trap temperature, °C	250
	Carrier gas flow rate (desorption from the trap), mL/min	50
	Desorption time, min	5
	Speed of the trap heating, °C/min	2000
GC-MS	Carrier gas	Helium
	Injector temperature, °C	250
	Split ratio	1:10
	Ion source temperature, °C	200
	Transfer line temperature, °C	250
	Scan mode	Full scan
	Scan range, amu	29–250
	Electron impact ionization, eV	70

**Table 3 metabolites-13-00203-t003:** Frequency of VOCs occurring in exhaled breath of healthy volunteers and lung cancer patients (%).

No	Retention Time, min	VOC	CAS Number	Lung Cancer Patients	Healthy Volunteers
1	7.40	Isoprene	78-79-5	100	100
2	6.20	Acetone	67-64-1	100	100
3	6.56	Dimethylsulfide	75-18-3	100	100
4	11.99	1-Methylthiopropene	10152-77-9	100	92
5	11.34	Allyl methyl sulfide	10152-76-8	98	97
6	11.76	1-Methylthiopropane	3877-15-4	98	95
7	11.43	2-Pentanone	107-87-9	97	99
8	12.20	Dimethyl disulfide	624-92-0	97	93
9	8.81	2.3-Butandione	431-03-8	88	92
10	5.20	Acetonitrile	75-05-8	87	88
11	8.95	2-Butanone	78-93-3	76	86
12	18.51	Dimethyl trisulfide	3658-80-8	71	55
13	18.26	Benzaldehyde	100-52-7	58	57
14	12.83	1-Pentanol	71-41-0	56	59
15	16.72	2-Heptanone	110-43-0	54	51
16	12.41	Heptane	142-82-5	53	73
17	22.45	Nonanal	124-19-6	53	56
18	9.86	Hexane	110-54-3	51	62
19	16.54	3-Heptanone	106-35-4	51	52
20	19.64	Octanal	124-13-0	51	52
21	15.13	Octane	111-65-9	51	50
22	13.36	Toluene	108-88-3	51	32
23	11.39	Pentanal	110-62-3	50	72
24	14.31	Hexanal	66-25-1	49	49
25	20.16	Decane	124-18-5	49	48
26	25.04	Dodecane	112-40-3	46	49
27	22.54	Undecane	1120-21-4	45	49
28	17.25	Propylbenzene	103-65-1	45	13
29	24.97	Decanal	112-31-2	42	49
30	17.15	Heptanal	111-71-7	39	48
31	8.90	Butanal	123-72-8	37	48
32	20.37	Nonane	111-84-2	37	46
33	10.44	Benzene	71-43-2	31	22
34	7.87	1.3-pentadiene	504-60-9	25	15
35	16.14	Ethylbenzene	100-41-4	25	12
36	10.32	1-Butanol	71-36-3	24	49
37	7.78	1.4-pentadiene	591-93-5	22	10
38	14.57	Butyl acetate	123-86-4	21	48
39	16.25	o-Xylene	95-47-6	20	13
40	16.76	M + p-Xylene	108-38-3 + 106-42-3	20	12

**Table 4 metabolites-13-00203-t004:** Correlation coefficients between VOCs (their ratios) and TNM stage.

VOC (Ratio)	Correlation Coefficient
2,3-Butandione	0.343
Dimethyl trisulfide	−0.235
Octane	0.272
Octane/Acetone	0.300
Octane/Acetonitrile	0.319
Octane/Isoprene	0.312
Octane/1-methylthiopropene	0.356
Octane/Dimethyl disulfide	0.375
Dimethyl trisulfide/Acetone	−0.272
Dimethyl trisulfide/Isoprene	−0.237
Dimethyl trisulfide/2-butanone	−0.259
Dimethyl trisulfide/dimethyl sulfide	−0.242
Dimethyl trisulfide/dimethyl disulfide	−0.237
Dimethyl trisulfide/2-pentanone	−0.279
Benzaldehyde/acetonitrile	0.249
2,3-Butandione/2-pentanone	0.380
Acetonitrile/allyl methyl sulfide	0.275

**Table 5 metabolites-13-00203-t005:** Correlation coefficients between VOCs (their ratios) and histological type.

VOC (Ratio)	Correlation Coefficient
Octane/acetone	0.207
3-Heptanone/acetone	0.234
3-Heptanone/2-butanone	0.229
3-Heptanone/allyl methyl sulfide	0.229
3-Heptanone/1-methylthiopropane	0.235
Dimethyl trisulfide/Acetone	0.204
Dimethyl trisulfide/Isoprene	0.199
Dimethyl trisulfide/2-butanone	0.244
Dimethyl trisulfide/dimethyl sulfide	0.215
Dimethyl trisulfide/1-methylthiopropane	0.256

**Table 6 metabolites-13-00203-t006:** Ratios selected for the creation of diagnostic models.

Ratio	Correlation Coefficient		
	Disease Status	Age (Lung Cancer Patients)	Age (Healthy Volunteers)
Hexane/2-Pentanone	−0.309	0.005	−0.024
Toluene/Acetone	0.252	−0.169	−0.038
1-Pentanol/Acetone	0.136	−0.089	0.026
Pentanal/2-Pentanone	−0.346	0.059	−0.159
Dimethyl trisulfide/Dimethyl disulfide	0.271	0.028	0.005
Nonanal/2.3-Butandione	−0.153	0.082	−0.166
Heptane/Allyl methyl sulfide	−0.157	−0.028	0.135
2-Butanone/2-Pentanone	−0.320	0.006	−0.121
Isoprene/Acetone	0.227	−0.163	0.174
1-Methylthiopropane/Acetone	−0.149	−0.146	0.175
Dimethyl sulfide/Acetone	0.205	−0.123	0.097
Acetonitrile/Acetone	−0.269	−0.033	0.149

**Table 7 metabolites-13-00203-t007:** Accuracy of diagnostic models.

Machine Learning Algorithm	Training Dataset						Test Dataset					
	Sensitivity, %			Specificity, %			Sensitivity, %			Specificity, %		
Dataset	1	2	3	1	2	3	1	2	3	1	2	3
GBDT	92	94	96	82	85	92	88	78	77	77	68	81
ANN	89	88	87	85	85	75	88	85	82	86	80	81

## Data Availability

Data available on request due to restrictions eg privacy or ethical. The data presented in this study are available on request from the corresponding author. The data are not publicly available due to [ethical restrictions].

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
