# Peer review of "Non-Invasive Lung Cancer Diagnostics through Metabolites in Exhaled Breath: Influence of the Disease Variability and Comorbidities"

_metabolites, 2023, doi:10.3390/metabo13020203_

Round 1
Reviewer 1 Report
In the manuscript non-invasive method for lung cancer diagnostics in exhaled breath.has been described. The influence of the disease variability and comorbidities was also considered. The on markers for diagnostic of various disease are very important scientific aspects. This is especially important in the case of diseases such as cancers, where early diagnosis of the disease significantly increases the chances of its successful treatment. The development of non-invasive diagnostic methods is also very beneficial for patients and could be applied both to the early diagnosis of this disease and in monitoring the process of its treatment The manuscript are clearly described, obtained results are summarized in corresponding Tables and Figures. The obtained results were extensively compared with those previously published.
However, numerous investigations on the diagnosis of lung cancer in exhaled breath have been described in the literature. For this reason, the advantages and especially novelty of the described results should be more clearly emphasized. Conclusions from the conducted research should be also clearly formulated.
Author Response
Reviewer 1. In the manuscript non-invasive method for lung cancer diagnostics in exhaled breath.has been described. The influence of the disease variability and comorbidities was also considered. The on markers for diagnostic of various disease are very important scientific aspects. This is especially important in the case of diseases such as cancers, where early diagnosis of the disease significantly increases the chances of its successful treatment. The development of non-invasive diagnostic methods is also very beneficial for patients and could be applied both to the early diagnosis of this disease and in monitoring the process of its treatment The manuscript are clearly described, obtained results are summarized in corresponding Tables and Figures. The obtained results were extensively compared with those previously published.
However, numerous investigations on the diagnosis of lung cancer in exhaled breath have been described in the literature. For this reason, the advantages and especially novelty of the described results should be more clearly emphasized. Conclusions from the conducted research should be also clearly formulated.
Response to reviewer 1.
Thank you for the comment. We modified the end of introduction section and added conclusion section, where more clearly emphasized advantages, novelty and conclusion.
We modified introduction section as follows: line 84-102: «This research is devoted to the study of lung cancer variability and comorbidities. Ef-fect of chronic heart failure, hypertension, anemia, acute cerebrovascular accident, obesity, diabetes, on VOC profile of lung cancer patients were studied. The influence of most fre-quently occurring comorbidities among lung cancer patients were investigated for the first time. TD-GC-MS was used to analyze exhaled breath samples of lung cancer patients and healthy volunteers. VOC peak areas and their ratios were considered as quantitative pa-rameters. Additionaly, variation of exhaled breath composition of lung cancer patients dependently from tumor histological type, localization, TNM stage, effect of chemotherapy was studied. The parameters, which varied dependently from treatment status and comorbidities, were not considered as putative lung cancer biomarkers. Two kinds of machine learning algorithms, gradient boosted decision trees (GBDT) and artificial neural network (ANN) were applied for the creation of a diagnostic model».
The following conclusion section was added: line 412-421:
«5. Conclusions
The study shows which not only pulmonary, but non-pulmonary comorbidities can influence the exhaled breath VOC profile. Chronic heart failure and hypertension effect mostly. Variations of exhaled breath VOC profiles among lung cancer patients with dif-ferent histological types, TNM stages, tumor localization, TNM stages and treatment sta-tus are observed, which can influence on performance of diagnostic models. These factors should be considered before creating of lung cancer diagnostic model, which allows to build useful test for helping putative lung cancer patients to diagnose the dangerous dis-ease in time.»
The manuscript was proofread by a special in English editing.

Reviewer 2 Report
The manuscript entitled „Non-Invasive Lung Cancer Diagnostics through Methabolites in Exhaled Breath Influence of the Disease Variability and Comorbidities” is a valuable study in the field of metabolomic profiling. The work describes the results of exhaled breath analysis of 112 lung cancer 13 patients and 120 healthy individuals using gas chromatography-mass spectrometry (GC-MS).
The structure of the manuscript is correct, the methods are described properly and results are presented in the correct form.
Some minor errors are as follows:
1) 141 please write "2" as superscript in "m2/g" abbreviation;
2) 196 references to figures and tables in the text should start with a capital letter, e.g. (Fig. 1)
3) 199 I'm not sure which chromatogram (upper or lower) corresponds to patients/control. Please fix it.
4) It might be a good idea to list the retention times of the substance in Table 3
The work is written honestly. The research is within the scope of the chosen special issue., so I recommend this manuscript for publication in Metabolites Journal.
Author Response
Reviewer 2. The manuscript entitled „Non-Invasive Lung Cancer Diagnostics through Methabolites in Exhaled Breath Influence of the Disease Variability and Comorbidities” is a valuable study in the field of metabolomic profiling. The work describes the results of exhaled breath analysis of 112 lung cancer 13 patients and 120 healthy individuals using gas chromatography-mass spectrometry (GC-MS).
The structure of the manuscript is correct, the methods are described properly and results are presented in the correct form.
Some minor errors are as follows:
1) 141 please write "2" as superscript in "m2/g" abbreviation;
2) 196 references to figures and tables in the text should start with a capital letter, e.g. (Fig. 1)
3) 199 I'm not sure which chromatogram (upper or lower) corresponds to patients/control. Please fix it.
4) It might be a good idea to list the retention times of the substance in Table 3
The work is written honestly. The research is within the scope of the chosen special issue., so I recommend this manuscript for publication in Metabolites Journal.
Response to reviewer 2.
Author response 1. Thank you for the notice! We fixed the typo: «Sorbent tubes with the external diameter and length of 6.2 and 115 mm filled with 0.4 g of Tenax TA (60-80 mesh, Chromatec) sorbent with the surface area of 35 m2/g were used to preconcentrate VOCs».
Author response 2. Thank you for the notice! All references to figures and tables were issued according to the Journal’s requirements.
Author response 3. Thank you for the notice! We added the designations in Figure 1.
Author response 4. Thank you for the recommendation! The retention times of the substances were added in Table 3.
The manuscript was proofread by a special in English editing.

Reviewer 3 Report
In this study, authors used gas chromatography-mass spectrometry (GC-MS) to detect and compare metabolites in exhaled breath between lung cancer patients and healthy individuals. The results of this paper may provide help for lung cancer therapy in the future. I just have a few questions here.
1. Line 111, I do not think the title of the table is correct.
2. Table 1, There are large differences in the median value of age and the gender ratio between healthy people and patients.
3. Figure 2, did author have parallel experiments for each value, I did not see SD value in the bar figure.
4. Authors should emphasize how their study can benefit lung cancer patines.
Author Response
Reviewer 3. In this study, authors used gas chromatography-mass spectrometry (GC-MS) to detect and compare metabolites in exhaled breath between lung cancer patients and healthy individuals. The results of this paper may provide help for lung cancer therapy in the future. I just have a few questions here.
- Line 111, I do not think the title of the table is correct.
- Table 1, There are large differences in the median value of age and the gender ratio between healthy people and patients.
- Figure 2, did author have parallel experiments for each value, I did not see SD value in the bar figure.
- Authors should emphasize how their study can benefit lung cancer patines.
Response to Reviewer 3.
Author response 1. Thank you for the notice! The table was entitled as follows: Line 111: «Table 1. Clinical characteristics of subjects».
Author response 2. Thank you for the comment! In this study, most of healthy volunteers were significantly younger than lung cancer patients, which is one of the limitations of the study. However, the range of ages in both groups of participants was within 21-67(or 77) years old. Moreover, is should be noted that the young participants (from 21 years old) were also present in the group of lung cancer patients. Unfortunately, the increasing trend in the number of young people among cancer patients should be taken into account. To avoid the influence of age on the performance of the diagnostic model, we conducted correlation analysis between the putative biomarker peak area and age in each group (line 286, table 6). As it can be seen, all parameters selected for the creation of diagnostic models were not correlated with age in both groups, which proves the lack of age influence on the results.
Author response 3. Figure 2 represents the importance of each variable in diagnostic model which was built using GBoost. The dataset was divided into 3 parts. Three models were built and each part of dataset was used as the test dataset. Thus, all the data was used as the test dataset which shows the results fully. Parallel experiments for each value were not conducted.
Author response 4. Thank you for the notice! We added the conclusion section and emphasized the significance of our study for patients with lung cancer.
The manuscript was proofread by a special in English editing.
